# Towards Exciton-Polaritons in an Individual MoS_2_ Nanotube

**DOI:** 10.3390/nano10020373

**Published:** 2020-02-20

**Authors:** Dmitrii Kazanov, Maxim Rakhlin, Alexander Poshakinskiy, Tatiana Shubina

**Affiliations:** Ioffe Institute, 26 Politekhnicheskaya, 194021 St Petersburg, Russia; maximrakhlin@mail.ru (M.R.); poshakinskiy@mail.ioffe.ru (A.P.); shubina@beam.ioffe.ru (T.S.)

**Keywords:** nanotubes, monolayer, flake, WGM, micro-photoluminescence, exciton-polariton, 2D materials, TMDC, strong coupling

## Abstract

We measure low-temperature micro-photoluminescence spectra along a MoS${}_{2}$ nanotube, which exhibit the peaks of the optical whispering gallery modes below the exciton resonance. The energy fluctuation and width of these peaks are determined by the changes of the nanotube wall thickness and propagation of the optical modes along the nanotube axis, respectively. We demonstrate the potential of the high-quality nanotubes for realization of the strong coupling between exciton and optical modes when the Rabi splitting can reach 400 meV. We show how the formation of exciton-polaritons in such structures will be manifested in the micro-photoluminescence spectra and analyze the conditions needed to realize that.

## 1. Introduction

Nanotubes (NTs) made of transition metal dichalcogenides (TMD) such as MoS${}_{2}$, and WSe${}_{2}$, WS${}_{2}$ (generalized formula is MX${}_{2}$) were first synthesized in the last century [1,2] and have been intensively investigated since then (see for a review, [3]). The walls of the TMD NTs consist of monolayers connected by weak van der Waals forces [4]. In the last few years, TMD structures as a whole gained attention due to the exceptional optical properties in the monolayer limit. In particular, the MoS${}_{2}$ monolayer has the direct optical transitions in the visible range that are associated with the A-exciton, which has a large oscillator strength [5,6]. Recently, we have shown that the micro-photoluminescence (micro-PL) spectra of the multi-walled MoS${}_{2}$ NTs also exhibit the direct exciton emission [3]. Furthermore, the spectra are modulated by pronounced peaks linearly polarized along the tube axis that are attributable to the optical whispering gallery modes (WGMs) maintained inside the NT wall [7].

The presence of both strong exciton and optical resonances opens up a path to coupling them when their frequencies are close to each other. Then, the hybrid polariton modes are formed and the interaction strength can be quantified by the Rabi splitting between them. For planar microcavities based on classical semiconductors, the Rabi splitting $\hbar \Omega_{\mathrm{Rabi}}$ reaches values of about 10 meV for GaAs, 20 meV for CdTe, 50 meV for GaN, and 200 meV for ZnO [8,9,10,11]. Major interest in exciton-polaritons in 2D van-der-Waals structures is related to the enhanced interaction between light and matter [12] compared to classical bulk materials. In Fabry-Perót type microresonators with a single MoSe${}_{2}$ monolayer, the Rabi splitting $\hbar \Omega_{\mathrm{Rabi}}∼$ 20 meV was demonstrated. Placing *N* monolayers inside the cavity enhances the interaction by $\sqrt{N}$, and for four layers of MoSe${}_{2}$ the Rabi splitting of ∼40 meV [13] was achieved. A stronger interaction is realized for microcavities with WS${}_{2}$ monolayers where the Rabi splitting reaches 270 meV for A-exciton and 780 meV for B-exciton [14] at low temperatures, and is about 70 meV [15] at room temperature. Geometry more complex than planar allows for fine tuning of polariton spectra. The interaction of excitons with the waveguide mode in the structures based on MoSe${}_{2}$ was demonstrated to produce the Rabi splitting of ∼100 meV [16]. Possible Rabi splitting of about 280 meV was derived via analysis of extinction spectra of an ensemble of WS${}_{2}$ nanotubes [17].


In recent years, the TMD NTs have been actively studied for various applications. Reducing the crystal symmetry from the monolayer to the tubular structure can significantly increase the quantum efficiency for photovoltaic effect [18,19]. Hybrid structures, such as WS${}_{2}$-nanotube/graphene, demonstrate efficient photon absorption and optical carrier generation [20]. Individual TMD NTs were successfully used in nano-optoelectronics as detectors [21] and in nano-electronics as transistors [22]. It can be assumed that strong optical and exciton resonances in TMD nanotubes can lead to the formation of exciton-polaritons, which have many potential applications. However, little is known about the light–matter interaction in a nanotube.


In this paper, we study the radiation from an individual MoS${}_{2}$ nanotube with spatial resolution. We simulate micro-PL spectra and reproduce the peaks associated with WGMs. Our studies show that the position of the peaks varies along the NT due to a change in the wall thickness, while the peak broadening is determined by the dispersion of the optical modes which have a finite wave vector along the nanotube axis. While the structure we study operates in the weak coupling regime, we predict that in higher-quality nanotubes, the strong coupling between excitons and WGMs can be realized. We show how the emission spectra are transformed with the different degree of the coupling and analyze the conditions necessary to observe the exciton-polaritons in the TMD nanotubes.

## 2. Experiment and Modeling of Optical Modes

The nanotubes can be either synthesized [2] or rolled up, like in [23]. The synthesized TMD NTs are made by different methods, such as sulfurization, decomposition of precursor crystals, misfit rolling, direct synthesis from the vapor phase, and many others described in [24]. We studied nanotubes made by a chemical transport reaction at 1000 K using iodine as the transport agent in an evacuated silica tube at a pressure of 10${}^{-3}$ Pa and with a temperature gradient of 2K/cm [2]. This method enables us to create almost perfect NTs with a very low density of structural defects.

To study the optical properties of single MoS${}_{2}$ NTs, a micro-PL experiment was performed. A sample with NTs on a Si substrate [7] was mounted in a He-flow cryostat with an Attocube XYZ piezo-driver inside, which allowed us to precisely maintain the positioning of a chosen part on a NT with respect to a laser spot. Micro-PL measurements were performed at low temperature (10 K) when the direct exciton emission prevailed [3]. PL accumulation time was about 1 s. Focusing of a laser beam on the sample was carried out using a 50-fold objective (Mitutoyo 50xNIR, NA=0.42), which also was used to collect the PL signal. The enlarged image of the sample was transferred by means of achromatic lens to the plane of a mirror with a calibrated aperture (pinhole–200 μm). This arrangement determined the region on the sample from which the micro-PL signal was detected. To record the PL spectra from the NT, the collected signal passed through the gratings of the monochromator and entered the CCD camera. The spectral resolution of our micro-PL setup is 30 μeV in the visible range. A 405-nm line of a semiconductor laser was used for excitation. The laser power density corresponds to ∼10 mW per area of about 20–50 μm${}^{2}$.

Figure 1a shows a photo of a NT taken inside of the micro-PL setup. The NT is about 50 microns in length and 2 microns in diameter. The green dots p1–p8 show the locations from which the micro-PL signal was recorded. Their size corresponds approximately to the size of the signal detection spot in the experiment. In this experiment, we use the same nanotube which was investigated in [7], where the WGMs were first observed. To increase a common PL intensity, we exploit non-polarized excitation and non-polarized detection. As a result, the relative intensity of the WGM peaks, which are predominantly TM-polarized, is lower than previously shown. The electrodynamic calculations showed a complete coincidence of the experimental and theoretical values of the mode energies. The number of monolayers inside the nanotube wall was estimated as 45±5. This is well consistent with the preliminary TEM study of free-standing NT with a similar micro-PL spectrum, directly measured in a TEM grid as in [3].

The low-temperature micro-PL spectra measured in different spots of the NT are shown in Figure 2a. The black line indicates the spectrum taken from the surface of the plane layer (flake) featuring a bright, inhomogeneously broadened exciton resonant peak. The major peaks in the micro-PL spectra recorded from the NT are wider and red shifted by ∼50 meV with respect to the flake due to uncompensated stresses in the tube wall. The 3R-polytype stacking of the monolayers in chiral NTs can also contribute to this shift [25]. While the major peak at about ∼1.83 eV corresponds to A-exciton in MoS${}_{2}$, the series of smaller peaks at lower energies stems from optical WGMs. Here, they are characterized by angular quantum number m=18–21. Modes with energies higher than the exciton energy are not observed due to the strong absorption. The black dashed lines in Figure 2a indicate the maximum deviation in the positions of the peaks detected from different spot of the NT. For all of the observed modes, the spread is around 6 meV, which corresponds to a change of the wall thickness by one monolayer.

The intensity of the PL is maximal in the center of the NT, point p5 in Figure 1a, and decreases at the NT end. Analyzing the evolution of the spectra from point p5 to point p8 in Figure 2a, we can assume that the geometry of the NT is adiabatically changing as it becomes less like a NT and more like a ribbon (flattened tube). This is also confirmed by a decrease in the intensity of the peaks associated with the presence of optical modes and an increase in the intensity of the peak associated with the flat layer. It should be noted that NTs start to form inside microfolds or bend the edges of curved flakes [2]. To put a NT on the SiO${}_{2}$ substrate, one should tear it from the silica ampoules, where the NTs are grown. The point p8 corresponds to the tip of the NT, which consists of a chunk of the flake and the incipient part of the NT. Thus, due to the limited spatial resolution, we observe PL contributions from both the NT and the flake, which have approximately the same thickness. The other end of the tube (points p2–p4) turns out to be undamaged, as we observe pronounced optical modes in the corresponding spectra. In addition, we show the TM-polarized micro-PL spectrum measured at low temperatures where the peaks corresponding to the WGMs are more pronounced (see the inset in Figure 2b).

To model the PL spectra, we solve Maxwell’s equations for a hollow cylinder with certain inner and outer radii, dielectric permittivity of MoS${}_{2}$, and homogeneously distributed sources of radiation. We neglect the reduction of the tube symmetry due to the presence of the SiO${}_{2}$ substrate since the corresponding splittings are not observed in the experiment. Generalizing the approach developed in [7], we calculate the radiation intensity not only in the direction perpendicular to the NT axis, but also in other directions, characterized by angle θ; see Figure 1b. In the experiment, the PL from the directions with $\theta ≲25^{\circ }$ is collected by the microscope objective. Therefore, the PL spectra are contributed by the optical modes of the frequency ω corresponding wave vector along the tube axis K=(ω/c)sinθ≲3.5μm${}^{-1}$ (*c* is the speed of light). The spread of their energies widens the PL peaks.

Figure 2b shows the dependence of the energies of the optical modes with angular numbers m= 18–21 on the wave vector along the NT axis *K*. Calculations were made for NTs with the wall consisting of 44 (dashed lines) and 46 monolayers (solid lines). Reduction of the NT wall thickness leads to the increase of the mode energies due to the stronger confinement. The dependence of the mode energies on the wave vector *K* can be expressed as $E\left(K\right)=\hbar c/n_{\mathrm{eff}}\sqrt{K^{2}+{\left(m/R\right)}^{2}}$, where *R* is the NT radius; and $n_{\mathrm{eff}}$ is the effective refractive index, which can be estimated as $n_{\mathrm{eff}}\approx n\eta$ with η≈0.44 being the part of the electric field that is confined inside the NT wall. For small wave vectors K≪m/R considered here, the dispersion is quadratic, $E\left(K\right)=E\left(0\right)+\hbar^{2}K^{2}/2M^{*}$, with the effective mass $M^{*}=\hbar mn_{\mathrm{eff}}/\left(Rc\right)\approx 7.7\times 10^{-6}m_{0}$, where $m_{0}$ is the mass of the free electron. The numerically calculated dispersion shown in Figure 2b yields the close value $M^{*}=7.4\times 10^{-6}m_{0}$. Then, the PL peak broadening due to finite spread of detection angles is estimated as $\delta E\approx E^{2}\sin^{2}\theta /\left(2M^{*}c^{2}\right)=55\;$ meV. This value shows the upper bound of the peak width observed in the experiment.

To sum up all of the above, PL spectra indicate that the actual tube of MoS${}_{2}$ is not homogeneous. The observed variation of the peak positions in the PL spectra from different parts of the NT is due to the fluctuations of the NT wall thickness. The PL peak width is additionally influenced by the finite detection angle because of the optical modes dispersion along the tube axis. Higher homogeneity of NTs and a decrease of the detection angle could further sharpen the peaks. Then one could expect transition to the strong light-matter coupling regime with the formation of exciton-polaritons. Below, we discuss the signatures of such a transition and analyze the conditions necessary for their observation.

## 3. Polaritons in a Nanotube Resonator

In the above-discussed experiment, the exciton resonance in the tube certainly has strong inhomogeneous broadening that prevents observation of exciton-polariton. Qualitatively, new effects are expected in the higher-quality structures with sharp exciton resonances where the strong coupling regime between the optical modes of the NT and the excitons can be realized. The important role is played by the wave vector along the NT axis, *K*, that allows for the fine tuning of the optical mode energy to match the exciton energy. Here, we discuss the properties of such hybrid exciton-polariton modes and show how they would be manifested in the PL spectra.

To calculate the dispersion of exciton-polaritons in the NT resonator, we assume that its walls are characterized by the local single-pole dielectric function
(1)$$\varepsilon \left(\omega \right)=\varepsilon_{b}\left(1+\frac{\omega_{\mathrm{LT}}}{\omega_{\mathrm{ext}}-\omega -i\Gamma }\right),$$
where $\omega_{\mathrm{ext}}$ is the exciton resonance frequency, Γ is the exciton non-radiative decay rate, and $\omega_{\mathrm{LT}}$ is an effective longitudinal-transverse splitting that characterizes the exciton-light coupling strength. We calculate the latter as [26] (2)$$\omega_{\mathrm{LT}}=\frac{2\Gamma_{0}c}{\omega_{\mathrm{exc}}d\sqrt{\varepsilon_{b}}},$$ where $1/(2\Gamma_{0})=0.23$ ps is the radiative exciton lifetime in the MoS${}_{2}$ monolayer [27], $\varepsilon_{b}=16.2$ is the background dielectric constant of bulk MoS${}_{2}$, and d=6.7 Å is the interlayer distance, which yields $\hbar \omega_{\mathrm{LT}}∼114$ meV. We note that this value is valid for an array of isolated monolayers and shall be quenched if their interaction takes place. The interaction can be suppressed by considering nanotubes made of not densely packed monolayers, which can be both synthesized and rolled-up [23]. The real structures can have larger exciton radiative decay times. Table 1 below shows the values taken from the literature for MoS${}_{2}$ structures: monolayer, flake, and nanotube. We should note that some experiments faced the problem of the limited experimental resolution; thus, the measured values of decay give only the upper limits of the respective radiative lifetimes.

The strength of the interaction of a certain optical mode and exciton can be quantified by the value of the Rabi splitting $\hbar \Omega_{\mathrm{Rabi}}$ between the energies of the hybrid polariton modes, which are formed when the energies of the bare excitations, $E_{m}\left(K\right)$ and $E_{\mathrm{exc}}$ are close. Taking into account that only the fraction η≈0.44 of the optical mode has the electric field inside the NT wall and can interact with excitons (see [28] for a similar consideration for polaritons in cylindrical cavities), the Rabi frequency is calculated as
(3)$$\Omega_{\mathrm{Rabi}}=\sqrt{2\omega_{\mathrm{ext}}\omega_{\mathrm{LT}}\eta },$$
which yields $\hbar \Omega_{\mathrm{Rabi}}∼$ 400 meV. The Rabi splitting in the considered tubular structure surpasses the typical values for the resonators based on classical semiconductors, such as GaAs or GaN. This highlights the potential of the high-quality NT resonators based on MoS${}_{2}$ for realization of strong light-exciton interaction. In the state-of-the-art nanotubes, a potentially large Rabi splitting does not occur due to the strong inhomogeneous broadening.

To show the effect of a light-exciton interaction, we calculate the PL spectra each as a function of detection angle for different values of the Rabi splitting; see Figure 3. In the regime of weak interactions (see panel (a) for $\hbar \Omega_{\mathrm{Rabi}}=2$ meV) the PL spectra consists of the peaks corresponding to the optical modes with different angular numbers *m*. The intensities of the peaks grow when they approach exciton resonance energy, but their energies (dashed lines) remain unperturbed. The increase of the light-exciton interaction strength leads to appearance of a series of anti-crossings, each of them reflecting the interaction of an optical mode and the excitonic mode with the same angular quantum number *m*; see panels (b) and (c) calculated for $\hbar \Omega_{\mathrm{Rabi}}=20$ and 200 meV, corresponding to the variation of the radiative lifetime in Table 1. The modes with different *m*, which were degenerate in the absence of interaction, are now split into upper and lower polariton modes, and a broad peak at the exciton frequency appears between them. Therefore, the first signature of exciton-polaritons formation is the widening of the major PL peak.

## 4. Discussion and Outlook


We have predicted the possibility of strong coupling between light and excitons in high-quality NTs with low inhomogeneous broadening. However, the condition of high quality is necessary but not the only one that needs to be taken into account. The WGM peaks can appear either in the spectral range of indirect exciton, which is useless for polaritonic application, or can extend up to the range of a direct exciton, which can have strong oscillator strength even at room temperatures (see Figure 4a). This is because only the modes with the azimuthal angular numbers of m≃ 20 possess sufficient quality (Q) factor. Our calculations (Figure 4b) show that such WGMs definitely appear in the spectral range of interest when the tube has a radius around 1 μm and a wall width more than 40 monolayers. The nanotubes with smaller radii, even if they have thicker walls, are characterized by lower Q-factor. Indeed, we never observed the WGM peaks in the spectra of such NTs in our experiments.



In accordance with the world practice, the most effective studies of strong coupling need low temperatures, when the exciton resonances are most pronounced. Our experiments with individual NTs were done at 10 K as well. An additional benefit of that is the full suppression of indirect exciton emission—a parasitic channel of recombination (see the inset in Figure 2b). This phenomenon is suggestive of a thermally-activated energy transfer mechanism. The indirect exciton band appears with temperature and can dominate over the radiation of direct exciton (black line in Figure 4a). However, in some NTs, presumably not densely-packed, the direct exciton is rather pronounced up to room temperature and spectra have the WGM peaks (red line in Figure 4a). One can expect similar effects in the rolled-up nanotubes reported recently [23]. In this case, if the inhomogeneous broadening is low, the temperature of exciton-polariton observation can be increased.


Because the Q-factor of a tubular resonator depends on its geometry (diameter and wall), the application of TMD nanotubes for polaritonics needs their selection. Fortunately, it can be easily done by micro-PL spectroscopy in combination with modeling when specifying the number of monolayers is needed [7]. Besides, the polarization-sensitive measurements can be used to facilitate the observation of exciton-polaritons which predominantly arise from coupling with TM-polarized optical mode. Our approach can be generalized to describe formation of phonon-polaritons, observed in hBN nanotubes [32].


As for the practical applications, one should consider two cases of interaction: weak and strong coupling regimes. In the weak coupling regime such NTs could be used as polarization-sensitive photodetectors or filters, as in carbon nanotubes [33]. In the strong coupling regime, exciton-polaritons are one of the kind objects for the future polariton laser proposed in [34]. Moreover, such NTs form 1D polaritons and can be easily manipulated for creating photonic crystals, such as dynamic acoustic lattices and others [35,36]. Thus, with precise spatial and structural control of NTs, rapid progress will be made in highly potential nanosystems with enhanced optical and electronic properties.


Summarizing, we have studied the spatially-resolved luminescence along the MoS${}_{2}$ NTs. The measured spectra contain the peaks, whose energies depend on the monolayer fluctuation of wall width, while their broadening depends on a wave vector of excited optical modes. We have suggested that the high-quality nanotubes can be used as microcavities, promising efficient coupling between the optical modes and exciton resonances. We have calculated the PL spectra characteristic for weak and strong coupling regimes and discussed the conditions which will allow for experimental observation of exciton-polaritons in NTs. The unique coexistence of optical and exciton resonances in the TMD NTs could lead to large number of varying applications.

## Figures and Tables

**Figure 1 nanomaterials-10-00373-f001:**
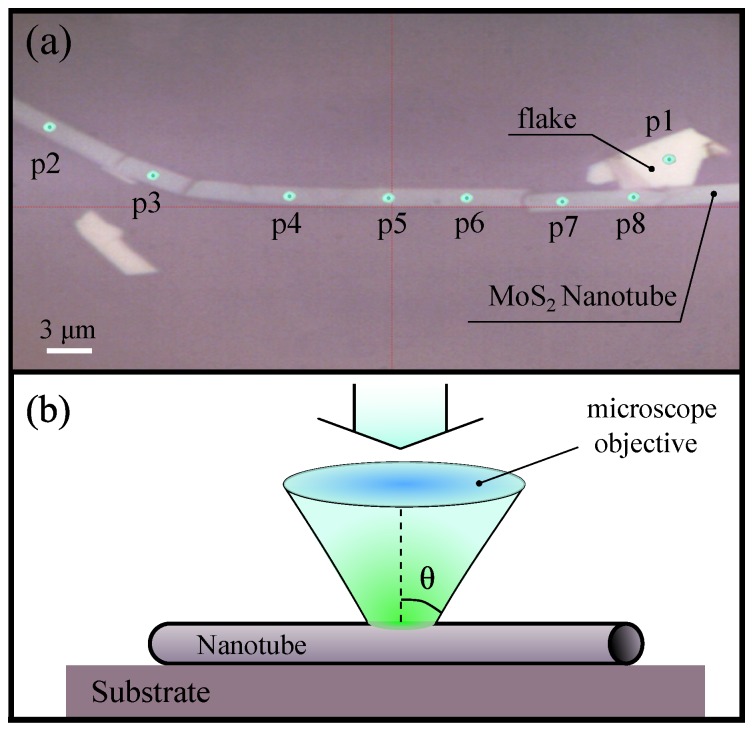
(**a**) Image of a MoS${}_{2}$ nanotube (NT) taken inside the micro-photoluminescence (micro-PL) setup. The NT has few cracks that are not essential for our experiment. Green dots indicate spots where the micro-PL was measured. (**b**) Schematic of NT excitation at micro-PL measurement (not to scale). As shown, the lens collects the emission at angles θ≲25.

**Figure 2 nanomaterials-10-00373-f002:**
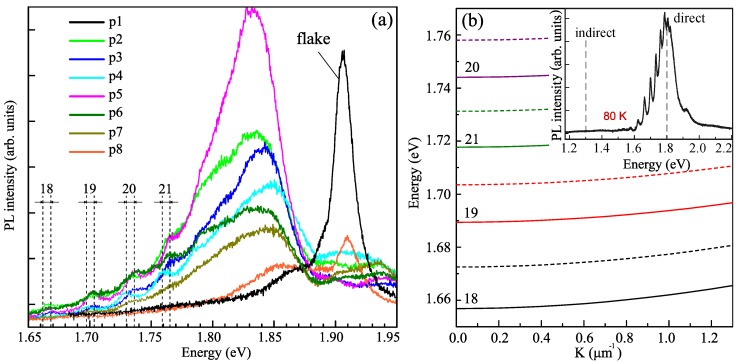
(**a**) Non-polarized micro-PL spectra measured at 10 K in the spectral range of direct exciton transitions from several spots of MoS${}_{2}$ NT, indicated by green dots in Figure 1a. Black line is a spectrum taken from the planar MoS${}_{2}$ layer (p1, flake). The dotted lines indicate the energies of optical modes with angular quantum numbers *m* = 18–21 calculated for the NT with the wall thickness varying by one monolayer. (**b**) Dependence of the energies of the optical modes with the angular numbers *m* = 18–21 on the wave vector *K* along the NT axis calculated for NTs with the 44-monolayer wall (dashed lines) and 46-monolayer wall (solid lines). The inset shows the TM-polarized micro-PL spectrum of the NT at 80 K.

**Figure 3 nanomaterials-10-00373-f003:**

Color plot of the PL’s spectral dependence on the detection angle calculated for the NTs without inhomogeneous broadening of the exciton resonance and different values of Rabi splitting, 2 meV, 20 meV, and 200 meV (**a**–**c**). The dotted lines show the dispersion of the eigenmodes with the angular quantum numbers m=19,20, and 21. The black solid line indicates the energy of the exciton. UPs and LPs denote upper and lower polariton modes, respectively.

**Figure 4 nanomaterials-10-00373-f004:**
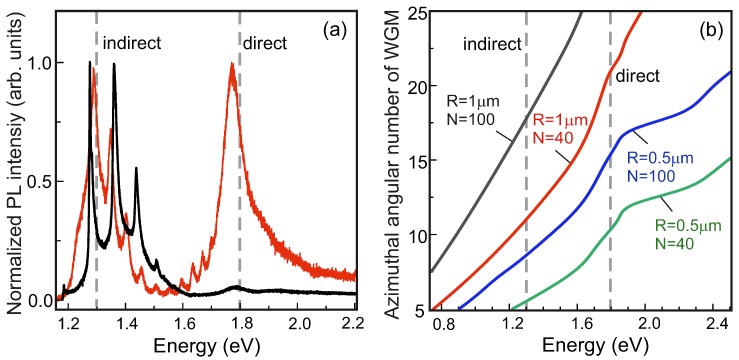
(**a**) TM-polarized micro-PL spectra measured in two different tubes of similar diameters (∼2 μm) at 300 K. (**b**) Azimuthal angular numbers of whispering gallery modes (WGM) in NTs vs. energy calculated for different radii, R, and numbers of monolayers, N.

**Table 1 nanomaterials-10-00373-t001:** Experimental and theoretical MoS${}_{2}$ exciton radiative lifetimes for different structures: monolayer, flake, and nanotube. Respective longitudinal-transverse splitting and Rabi splitting are calculated as described in the text.

MoS${}_{2}$ Structure	$\tau^{0}$, ps	$\hbar \omega_{\mathrm{LT}}$, meV	$\hbar \Omega_{\mathrm{Rabi}}$, meV
Monolayer, theory	0.23 [27]	114	400
Monolayer, experiment	4 [29,30,31]	6.5	98
Flake, experiment	30 [3]	0.86	35
Nanotube, experiment	26 [3]	1	38

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
