# Peer review of "Towards Exciton-Polaritons in an Individual MoS2 Nanotube"

_nanomaterials, 2020, doi:10.3390/nano10020373_

Round 1

Reviewer 1 Report

Authors studied the condition of exciton-polariton coupling in MoS2 nanotubes. PL spectra at low temperature were provided, which provided signatures of WGM and the energies of these modes were supported by the calculation. Exciton-polariton in TMD nanostructures such as nanotubes are important for the scientific interest and photonic applications. Devoted study of calculation and prediction could be beneficial to the communities of nanophotonics and TMD materials and therefore in principle I would recommend publication. However I’d like to raise following issues.

Experimental data of PL spectra from MoS2 nanotubes discussed here seem very similar to the previous APL work of the authors [ref. 7]. Is there unique information on experimental data given here different from the previous report? It is claimed that layers of N MoS2 monolayers are formed in MoS2 nanotubes. Isn’t it possible that nanotubes are just the rolled multilayers? How are the monolayer thickness confirmed here?

Reviewer 2 Report

The paper presents an interesting experiment and theoretically analyzes the propagation of polaritons in a TMDC nanotube geometry. The paper is interesting and timely. The authors correctly point out that their system does not show polariton due to the inhomogeneous broadening, but provide a theory to model them for hypothetical system that support them.
I believe the paper is suitable for publication, after the following two points have been addressed:

1)
What is the effect of the substrate on the model? I believe the presence of the substrate might break the rotational symmetry of the tube.

2)
How would your model compare with experimentally observed phonon polaritons in hexagonal boron nitride nanotubes (which have been observed experimentally with near field techniques)? The model and geometry should be similar (even though the physical effect creating the polariton is different), so you can potentially validate your model with those experimental observations. Please consider doing this comparison and adding your findings in the manuscript.

Round 2

Reviewer 1 Report

Authors addressed my concerns properly in the revision. I recommend the publication in its form.